# Two species of the green algae *Volvox* sect. *Volvox* from the Japanese ancient lake, Lake Biwa

Hisayoshi Nozaki[1,2,3]*, Ryo Matsuzaki[4], Koichi Shimotori[5], Noriko Ueki[6], Wirawan Heman[7], Wuttipong Mahakham[8], Haruyo Yamaguchi[2,5], Yuuhiko Tanabe[2], Masanobu Kawachi[2]

**1** Department of Biological Sciences, Graduate School of Science, The University of Tokyo, Tokyo, Japan, **2** Biodiversity Division, National Institute for Environmental Studies, Tsukuba, Ibaraki, Japan, **3** Department of Chemical and Biological Sciences, Faculty of Science, Japan Women's University, Tokyo, Japan, **4** Faculty of Engineering, Osaka Institute of Technology, Osaka, Japan, **5** Lake Biwa Branch Office, National Institute for Environmental Studies, Otsu, Shiga, Japan, **6** Science Research Center, Hosei University, Tokyo, Japan, **7** Department of Science and Mathematics, Faculty of Science and Health Technology, Kalasin University, Kalasin, Thailand, **8** Department of Biology, Faculty of Science, Khon Kaen University, Khon Kaen, Thailand

* hisayoshi.nozaki@gmail.com

**Data Availability Statement:** New sequence data of Volvox biwakoensis, V. kirkiorum and V. sp. Sagami are available under the DDBJ/ENA/ GenBank accession numbers (LCLC817957–

## Abstract

*Volvox* sect. *Volvox* is a group of green algae with unique morphological features (thick cytoplasmic bridges between somatic cells and spiny zygote walls) and a worldwide distribution. Despite research interest in the diversity of organisms in ancient lakes, *Volvox* sect. *Volvox* from ancient lakes worldwide has not been identified to the species level. Here, we established clonal cultures of two species of this group originating from Lake Biwa, an ancient lake in Japan, and performed identification based on morphological and molecular data. One was identified as *Volvox kirkiorum* based on the nuclear ribosomal DNA internal spacer region (ITS) sequence, bisexual (monoicous or monoecious) spheroids, and zygote morphology. The other showed genetic separation from related species based on the secondary structure of the ITS and results of phylogenetic analysis of a combined data set from the nuclear *actin* gene, ITS, and two plastid genes (large subunit of RuBisCO and photosystem II CP43 apoprotein gene); it represented a new phylogenetic lineage within *Volvox* sect. *Volvox*, suggesting possible endemism in Lake Biwa. This species produced bisexual spheroids with different zygote morphology and zygote number from other species with bisexual spheroids in *Volvox* sect. *Volvox*. Therefore, *Volvox biwakoensis* Nozaki et H. Yamaguchi sp. nov. is described herein. This is the first endemic species of the genus *Volvox* described from an ancient lake.

## Introduction

Ancient lakes are the oldest lakes in the world, dating from more than 1 million ago, and they have fascinated biologists with their high endemicity and diversity of animals, land plants, and

LC817993 and LC818981). All other relevant data are within the paper and its Supporting information files.

**Funding:** This study was partially supported by a Grants-in-Aid for Scientific Research (grant numbers 19K22446 and 24K08946 for HN, 21K06295 to NU) from the Ministry of Education, Culture, Sports, Science and Technology (MEXT)/ Japan Society for the Promotion of Science (JSPS) KAKENHI (https://www.jsps.go.jp/english/e-grants/ ), the Collaborative Research Fund (grant number OS2024RR1 to KS) from Shiga Prefecture entitled "Study on water quality and lake-bottom environment for protection of the soundness of water environment" under the Japanese Grant for Regional Revitalization (https://www.mlit.go.jp/en/ index.html), and the Research Program of "Dynamic Alliance for Open Innovation Bridging Human, Environment and Materials" (grand numbers 20211204, 20221238, 20231201 and 20241233 for NU) in "Network Joint Research Center for Materials and Devices" (https://alliance. tagen.tohoku.ac.jp/english/). The funders had no role in study design, data collection and analysis, decision to publish, or preparation of the manuscript.

**Competing interests:** The authors have declared that no competing interests exist.

algae [1–4]. Lake Biwa is an ancient lake that is one of the largest lakes in Japan. Aquatic land plants and aquatic multicellular animals in this lake have been studied extensively, and more than 60 endemic species and infraspecific taxa are known [5]. However, the volvocine green algae from Lake Biwa have been identified based only on light microscopic features of vegetative (asexual) stages [6], with the exception of our recent study of two species of *Volvox*, *Volvox africanus* and *Volvox reticuliferus* [7]. Although *V. reticuliferus* was described as a new species, it corresponds to the subdivision of *V. africanus* previously recorded worldwide [7, 8] and does not represent a new lineage or endemic species.

*Volvox* is a typical genus of volvocine green algae with multicellular spheroids, which is comprised of approximately 22 species worldwide [7, 9]. The genus is subdivided into four sections based on differences in vegetative and zygote morphology and phylogeny [7]. *Volvox* sect. *Volvox* is a unique group characterized by the presence of thick cytoplasmic bridges between the cells of the spheroids and spiny zygote walls [7]. Approximately 11 species have been described in this section worldwide [9]. However, two species of *Volvox* that we collected previously from Lake Biwa (*V. africanus* and *V. reticuliferus* [7]) belong to a different section, *Volvox* sect. *Merrillosphaera* [7]. Occurrences of the genus *Volvox* have been recorded in Lake Tahoe, an ancient lake in California, USA; however, no species identification has been described [10, 11]. *Volvox* sect. *Volvox* has not been identified to the species level from Lake Biwa, and there are no records of *Volvox* sect. *Volvox* from other ancient lakes worldwide [9, 12].

The present study was performed to identify species of *Volvox* sect. *Volvox* originating from Lake Biwa based on morphological and molecular data from cultured material. Here, we describe the morphology, molecular phylogeny, and taxonomy of two species of this section from Lake Biwa, *Volvox kirkiorum* Nozaki et al. and *Volvox biwakoensis* Nozaki et H. Yamaguchi sp. nov. representing a new endemic lineage within this section.

## Materials and methods

### Ethics statement

KS collected multicellular volvocine green algae from the water column of Lake Biwa in Japan. Collection of the samples from Lake Biwa was permitted by Biodiversity Strategy Promotion Office, Department of Environment of Lake Biwa, Shiga Prefecture, Japan. NU collected water samples of *Volvox* from a private boat dock area connected to Sotobori (outer moat), Shinjuku-ku, Tokyo, authorized by the owner (Canal Cafe <https://www.canalcafe.jp/>).

### Establishment of cultures and morphological observations

An offshore water sample was collected from Yanagasaki in the south basin (Nanko) of Lake Biwa (35.0261274, 135.8673386) on October 20, 2022. The surface area of the lake measures 670.25 km$^2$; its average and maximum depth are 41.2 m and 103.58 m, respectively; and the elevation of the surface of the lake above sea level is 84.371 m [13]. The water temperature at this site was 20.1˚C and pH was 8.0. An additional water sample was collected from the same site on October 26, 2022. Five clonal cultures of *Volvox* sect. *Volvox* (strains 2022-1021-Biwa1, 2022-1027-VVx8, 2022-1027-VVx9, 2022-1027-VVx11, and 2022-1027-VxA12) were established from these water samples using the pipette washing method [14] (Table 1). Cultures were grown in 18 × 150 mm screw-cap tubes containing 10–11 mL of artificial freshwater-6 (AF-6) medium [15] at 20˚C or 25˚C under a 14:10 h light:dark (L:D) cycle under cool-white fluorescent lamps with a color temperature of 5000 K at an intensity of 70–150 µmol m$^{-2}$ s$^{-1}$. To examine morphological details, possible bacterial contamination was removed from the cultures of 2022-1021-Biwa1, 2022-1027-VVx9, and 2022-1027-VVx11 by picking up a young

**Table 1. List of species/strains used in the present phylogenetic analyses (Figs 1 and 3).**

| Species | Sample/strain designation | Origin of sample/strain | GenBank/EMBL/DDBJ Accession number | | | |
|---|---|---|---|---|---|---|
| | | | ITS-1, 5.8S rDNA & ITS-2 | *actin* | *rbcL* | *psbC* |
| *Volvox biwakoensis* sp. nov. from Lake Biwa, Japan | 2022-1021-Biwa1 (= NIES-4661) [a] | Water sample collected from an offshore from Yanagasaki in the south area (Lake Nanko) of Lake Biwa (35.0261274, 135.8673386) on 20 October 2022 | LC817963 [b] | LC817974 [b] | LC817986 [b] | LC817992 [b] |
| *Volvox kirkiorum* | NIES-543 | Japan | AB663327 | LC817972 [b] | AB663325 | AB663326 |
| | NIES-2740 | Gifu, Japan | AB663324 | LC817973 [b] | AB663322 | AB663323 |
| | 2022-1027-VVx9 [a] | Water sample collected from an offshore from Yanagasaki in the south area (Lake Nanko) of Lake Biwa (35.0261274, 135.8673386) on 26 October 2022 | LC817959 [b] | LC817970 [b] | LC817984 [b] | LC817990 [b] |
| | 2022-1027-VVx11 [a] | | LC817960 [b] | LC817971 [b] | LC817985 [b] | LC817991 [b] |
| | 2022-1027-VVx8 [a] | | LC817961 [b] | | | |
| | 2022-1027-VxA12 [a] | | LC817962 [b] | | | |
| *Volvox* sp. Sagami | NIES-4021 | Kanagawa, Japan | LC191308 | LC818981 [b] | LC191316 | LC191326 |
| | NIES-4022 | | LC191309 | LC817975 [b] | LC191317 | LC191327 |
| | SB01 (= NIES-4662) [a] | A private boat embarkation area connected to Sotobori (outer moat), Shinjuku-ku, Tokyo (35.699994, 139.742759) on 15 May, 2021 | LC817964 [b] | LC817976 [b] | LC817987 [b] | LC817993 [b] |
| *Volvox longispiniferus* | 1101-NZ-5 (= NIES-4434) | Thailand | LC546057 | LC817979 [b] | LC546062 | LC546067 |
| *Volvox barberi* | UTEX 804 | USA | AB663341 | LC817977 [b] | D86835 | AB044477 |
| *Volvox capensis* | NIES-3874 | USA | LC034074 | LC817980 [b] | LC033870 | LC033872 |
| *Volvox ferrisii* | NIES-2736 | Kanagawa, Japan | AB663336 | LC817965 [b] | AB663334 | AB663335 |
| *Volvox globator* | SAG 199.80 (= UTEX 955) | USA | AB663340 | LC817981 [b] | D86836 | AB044478 |
| *Volvox perglobator* | NIES-4259 [c] | USA | | LC817978 [b] | | |
| | NIES-4258 [c] | | MG429137 | | KY489662 | KY489659 |
| *Volvox rousseletii* from South Africa | UTEX 1861 male | South Africa | LC817957 [b] | LC817967 [b] | LC817982 [b] | LC817988 [b] |
| | UTEX 1862 female | | AB663342 | LC817966 [b] | D63448 | AB044479 |
| *Volvox rousseletii* from Japan | NIES-4336 female | Japan | LC493797 | LC817968 [b] | LC493808 | LC493810 |
| | NIES-4337 male | | LC817958 [b] | LC817969 [b] | LC817983 [b] | LC817989 [b] |

[a] Established in this study.

[b] Sequenced in this study.

[c] Treated as a single strain for phylogenetic analysis of combined data set in Fig 2.

spheroid still within the parental spheroid and washing several times with fresh medium using a micropipette; the young spheroid was then transferred to 10–11 mL of urea soil *Volvox* thiamin (USVT)/3 medium [16] or *Volvox* thiamin acetate (VTAC) medium [15]. Asexual spheroids were observed in actively growing cultures with 10–11 mL or 30 mL of USVT/3 medium in screw-cap tubes or Petri dishes (20 × 90 mm), respectively, at 25°C under a 14:10 h L:D cycle. To induce sexual reproduction of *V. biwakoensis* strain 2022-1021-Biwa1, approximately 1 mL of actively growing culture in VTAC medium was inoculated with 10–11 mL of USVT medium in a screw cap tube. This culture was grown at 25°C under a 14:10 h L:D cycle. Sexual spheroids developed within 2 weeks. In *V. kirkiorum* strains 2022-1027-VVx9 and 2022-1027-VVx11, sexual spheroids often developed in actively growing cultures in USVT/3 medium at 25°C. To enhance formation and maturation of zygotes, immature sexual spheroids cultured under the conditions outlined above were transferred to fresh USVT/3 or USVT medium and cultured at 25°C under a 14:10 h L:D cycle.

A new clonal culture strain (SB01) of *Volvox* sp. Sagami (Table 1) was established from a water sample collected from a private boat dock area connected to Sotobori (outer moat), Shinjuku-ku, Tokyo (35.699994, 139.742759) on May 15, 2021. The water temperature at this site was 23.3˚C with a pH of 7.5. The methods used for this species were essentially the same as described above for *V. kirkiorum*.

Light microscopy was performed using a BX60 microscope (Olympus, Tokyo, Japan) equipped with Nomarski optics. The spheroid cell numbers were examined as described previously [12, 17]. Individual cellular sheaths of the extracellular matrix of the spheroids were examined after mixing approximately 10 μL of cultured material with 2–5 μL of 0.002% (w/v in distilled water) methylene blue (1B-429; Waldeck GmbH & Co., Münster, Germany).

### Molecular experiments

To infer the phylogenetic positions or species identity of the algae in *Volvox* sect. *Volvox*, we first determined the internal transcribed spacer (ITS) regions of nuclear ribosomal DNA (rDNA; ITS-1, 5.8S rDNA, and ITS-2) from the five strains of *Volvox* sect. *Volvox* from Lake Biwa (Table 1, S1 Appendix) as described previously [18], and performed comparisons with those of other strains of *Volvox* sect. *Volvox* studied previously [9] (Table 1). For a more robust determination of the phylogenetic positions of the Lake Biwa species, a combined data set from the ITS, *actin* gene, and two plastid genes [the large subunit of RuBisCO (*rbcL*) gene, plus the photosystem II CP43 apoprotein (*psbC*) gene] were constructed from 10 species of *Volvox* sect. *Volvox* listed in Table 1. The two plastid genes were determined as described previously [18, 19]. Nucleotide sequences of a potential group I intron inserted in the *Volvox* sp. Sagami *psbC* gene were determined by direct sequencing of PCR products amplified with four primers [16]. The *actin* genes were determined by the method of Kimbara et al. [18] with new primers (S1 Table). The ITS sequences in the two data sets were aligned as described previously [9, 19]. The *actin* exon–intron sequences from 13 operational taxonomic units (OTUs) of different sequences from 19 strains of the 10 species (Table 1) were aligned using MUSCLE [20] in MEGA X [21] and included in the data matrix (3192 bp, S2 Appendix). The outgroup or root was designated based on the results of previous phylogenetic analyses [9, 19]. Bayesian inference (BI) was performed with partitioned models as described previously [9]. In addition, maximum-likelihood (ML) analyses, based on the selected models (K80+G and T92+G models for ITS and the combined data set, respectively) with 1000 bootstrap replicates [22], were performed using MEGA X. The secondary structures of ITS-2 were predicted as described previously [8, 9].

### Nomenclature

The electronic version of this article in Portable Document Format (PDF) in a work with an ISSN or ISBN constitutes a published work according to the International Code of Nomenclature for algae, fungi, and plants [23]; hence, the new name contained in the electronic publication of a PLOS ONE article is effectively published under that Code from the electronic edition alone, so there is no longer any need to provide printed copies.

## Results and discussion

### Molecular analysis

Two different sequences were detected on examination of ITS from five strains of *Volvox* sect. *Volvox* from Lake Biwa. Strain 2022-1021-Biwa1 had a 590-bp ITS sequence (excluding the 5′ and 3′ primer sequences [17, 24]), while the other four had identical ITS sequences 558 bp in

length. Based on the results of phylogenetic analysis of ITS from various OTUs of *Volvox* sect. *Volvox* [9], strain 2022-1021-Biwa1 represented a novel lineage (*V. biwakoensis*) (Fig 1) whereas the other four had exactly the same sequence as *V. kirkiorum* [19]. However, the phylogenetic relationships between *V. biwakoensis* and its related taxa (*Volvox ferrisii*, *Volvox* sp. Sagami, *Volvox rousseletii*, and *V. kirkiorum*) were not well resolved (Fig 1).

The tree was constructed based on Bayesian Inference (BI) with K80+G model. Branch lengths represent the evolutionary distances shown by the scale bar. Posterior probabilities (0.90 or more) by BI and bootstrap values (50% or more) based on 1000 replications of maximum likelihood method (with K80+G model) are described in numbers in left and right sides at branches, respectively. For alignment data, see S1 Appendix.

Phylogenetic analyses of the combined data set from the nuclear gene *actin*, ITS, and two plastid genes (3192 bp) showed that *V. biwakoensis*, *V. rousseletii*, *V. ferrisii*, and *Volvox* sp. Sagami formed a monophyletic group with 0.99 posterior probability (PP) on BI and 83% bootstrap support on ML analysis (Fig 2). In this monophyletic group, *V. biwakoensis* was sister to the clade composed of the other three species with 0.94 PP on BI and 66% bootstrap support on ML analysis (Fig 2).

The tree was constructed based on Bayesian Inference (BI) with partitioned models (*rbcL*: HKY+I; *psbC*: F81+I; ITS: K80+G, *actin*: K80+I). Branch lengths represent the evolutionary distances shown by the scale bar. Posterior probabilities (0.90 or more) by BI and bootstrap

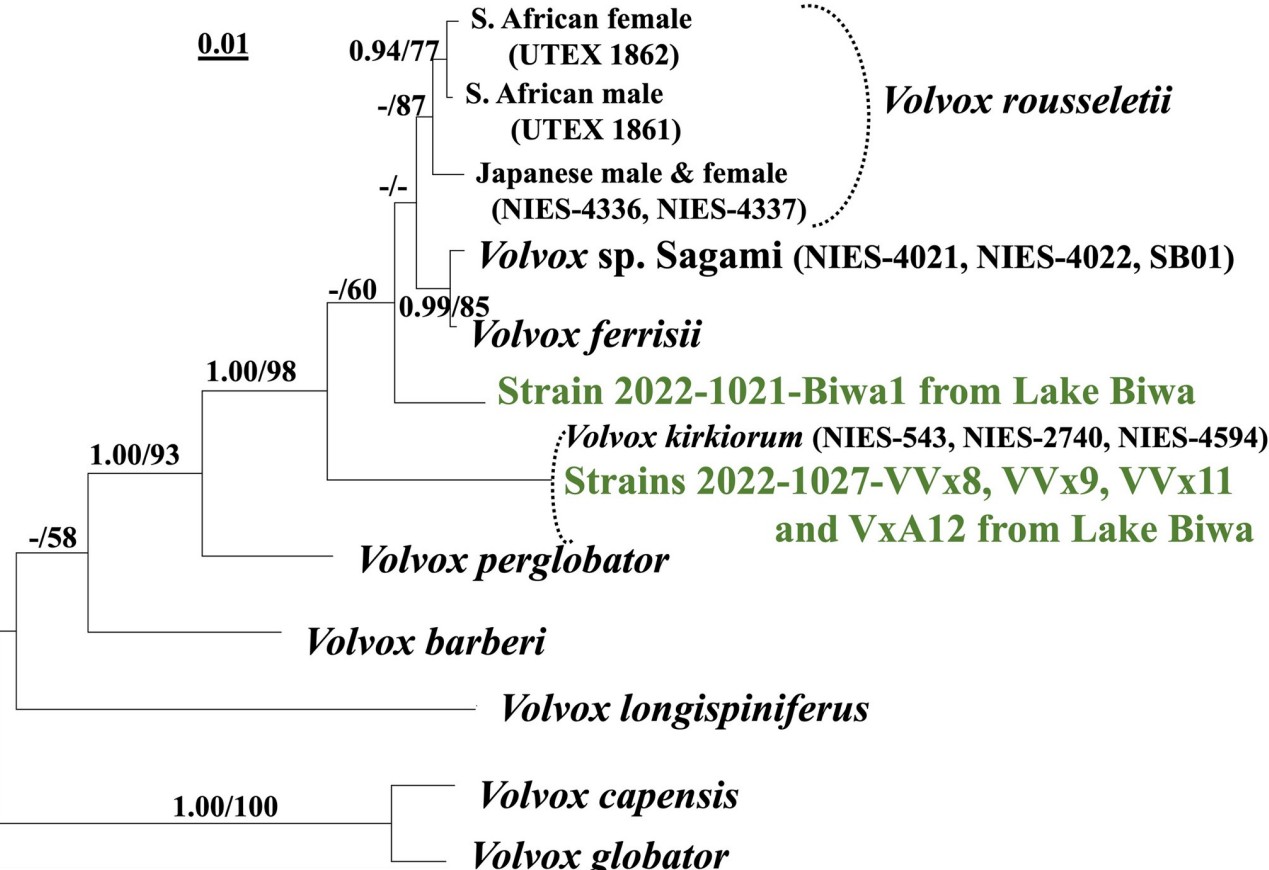

**Fig 1. Molecular identification of five new strains of *Volvox* sect. *Volvox* from Lake Biwa (green), based on the Internal Transcribed Spacer (ITS) regions of nuclear ribosomal DNA (rDNA) (ITS-1, 5.8S rDNA, and ITS-2) (Table 1).**

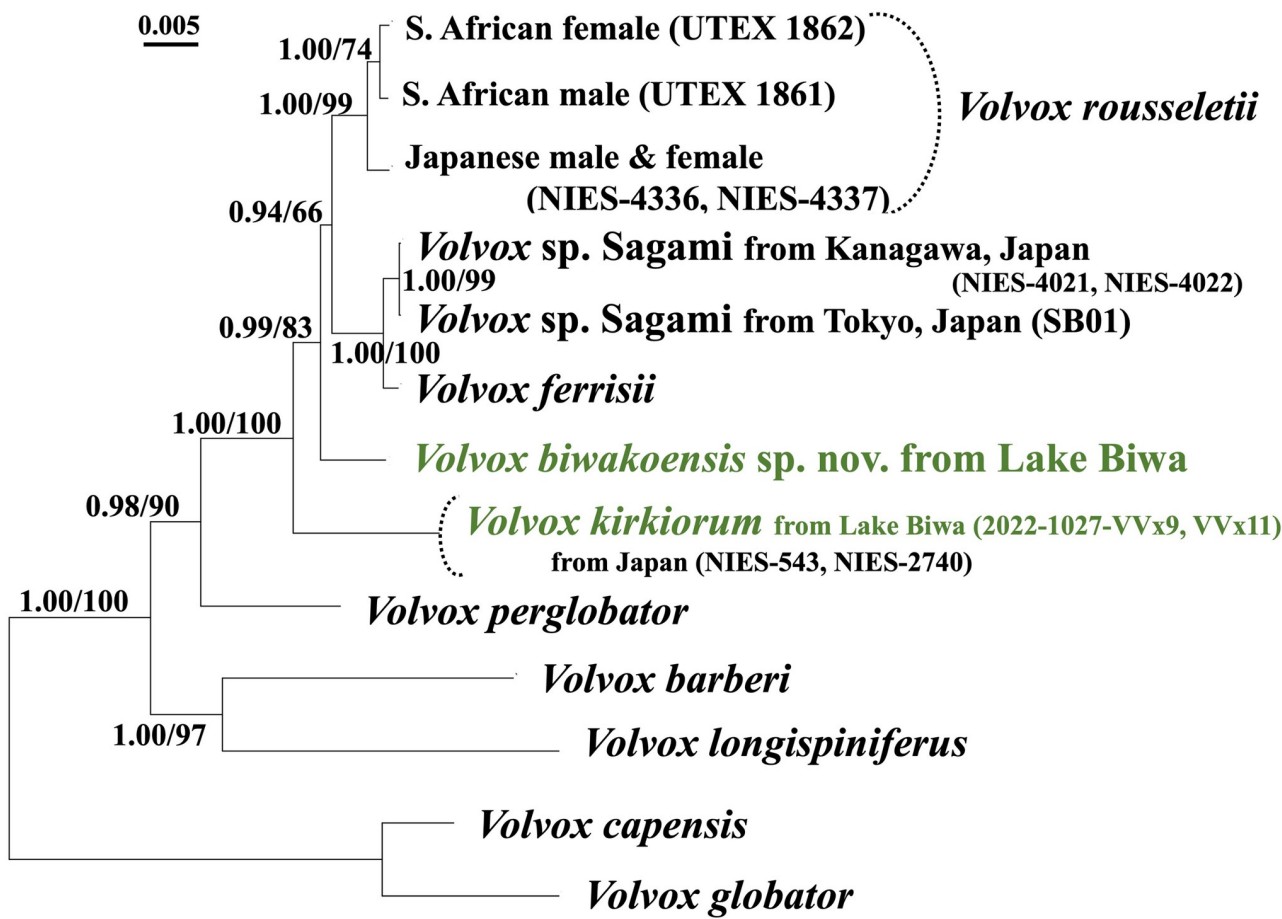

**Fig 2. Phylogenetic analysis of two species of *Volvox* sect. *Volvox* from Lake Biwa (green) based on combined data set from the Internal Transcribed Spacer (ITS) regions of nuclear ribosomal DNA (rDNA) (ITS-1, 5.8S rDNA, and ITS-2), *actin* gene and two chloroplast genes (*rbcL* and *psbC*) (Table 1).**

values (50% or more) based on 1000 replications of maximum likelihood method (with T92 +G) are described by numbers in left and right sides at branches, respectively. For alignment data, see S2 Appendix.

Based on comparisons of the secondary structure of ITS-2, one or two compensatory base changes (CBCs) were found between *V. biwakoensis* and its sister group (*V. rousseletii*, *V. ferrisii*, and *Volvox* sp. Sagami) (Fig 3).

Red frames indicate compensatory base changes between *V. biwakoensis* and other species. Nucleotide sequence of *V. rousseletii* represents mixed or common bases between Japanese (JP) and African (AF) strains (Table 1).

### *Volvox biwakoensis* sp. nov. (Fig 4)

Asexual spheroids were ovoid with a narrow anterior face, with 4–10 (usually 5–8) developing embryos or daughter spheroids in the posterior three-fifths of the parental spheroid, and were composed of 6000–9000 cells embedded in individual sheaths at the periphery of the gelatinous matrix (Fig 4A–4D). The mature spheroid measured up to 800 μm in length. Somatic cells were biflagellate and had a cup-shaped chloroplast with a single basal pyrenoid and a single eyespot,

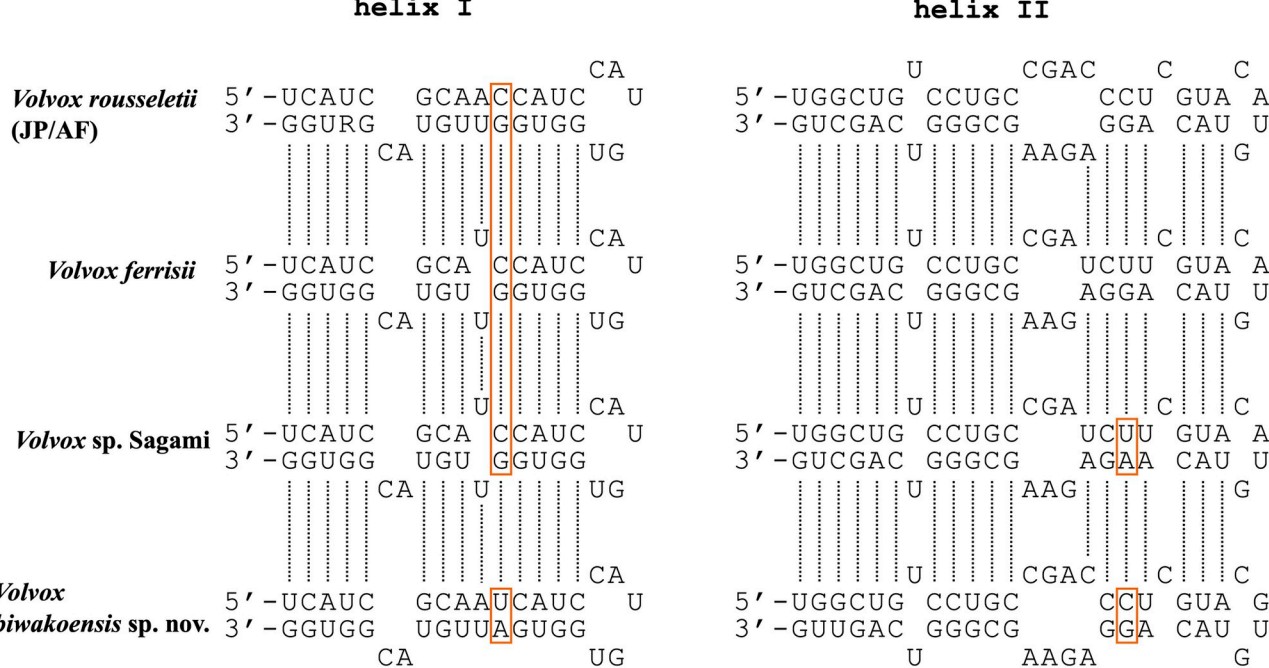

**Fig 3. Comparison of helices of the secondary structure of nuclear ribosomal DNA internal transcribed spacer 2 transcripts between *Volvox biwakoensis* sp. nov. and its related species (Fig 2).**

measuring up to 8 μm in length (Fig 4E and 4F). The cells were connected by cytoplasmic bridges that were thicker than the flagella (Fig 4D–4F). The anterior somatic cells of the spheroids were ovoid to ellipsoidal with a narrow anterior face, and the cell length was greater than or nearly equivalent to the cell width (Fig 4F). In the developing embryo, reproductive cells or gonidia of the next generation were not evident until the inversion stage (Fig 4G).

Sexual spheroids were bisexual or monoicous, subspherical or ovoid in shape, measured up to 650 μm in length, and consisted of 3600–5000 somatic cells at the periphery (Fig 4H). Four to eight sperm packets and four or more (usually 7–13) eggs were distributed in the posterior two-thirds of the spheroid (Fig 4H and 4I). Mature zygotes were 9–26 in number and reddish-brown in color with a spiny cell wall (Fig 4J). Fully developed spines of the zygote wall were 5–7 μm in length, straight or slightly curved, with an acute apex (Fig 4K). Zygotes were 46–50 μm in diameter, excluding spines. Induction of zygote germination was not attempted.

Species of *Volvox* sect. *Volvox* were subdivided into two morphological types based on the differences in sexual spheroids: bisexual (monoicous) or unisexual (dioicous) [9, 12]. *V. biwakoensis* belonged to the bisexual type and was similar to *Volvox* sp. Sagami (S1 Fig) in production of less than 30 zygotes with a maximum diameter of more than 45 μm (excluding spines) (S2 Table). However, the two species could be clearly distinguished from each other by differences in length of spines on the zygote wall and phylogenetic position. The spines of *V. biwakoensis* are 5–7 μm in length (Fig 4), while those of *Volvox* sp. Sagami were short, measuring up to 3 μm in length (S2 Table and S1 Fig). In the present phylogenetic analyses based on the combined data set from four DNA regions, *V. biwakoensis* was sister to the large clade composed of *V. rousseletii*, *V. ferrisii*, and *Volvox* sp. Sagami (Fig 2) and showed CBC in the ITS-2 secondary structure from its related species (Fig 3). Therefore, *V. biwakoensis* is an undescribed species representing a new lineage within *Volvox* sect. *Volvox*.

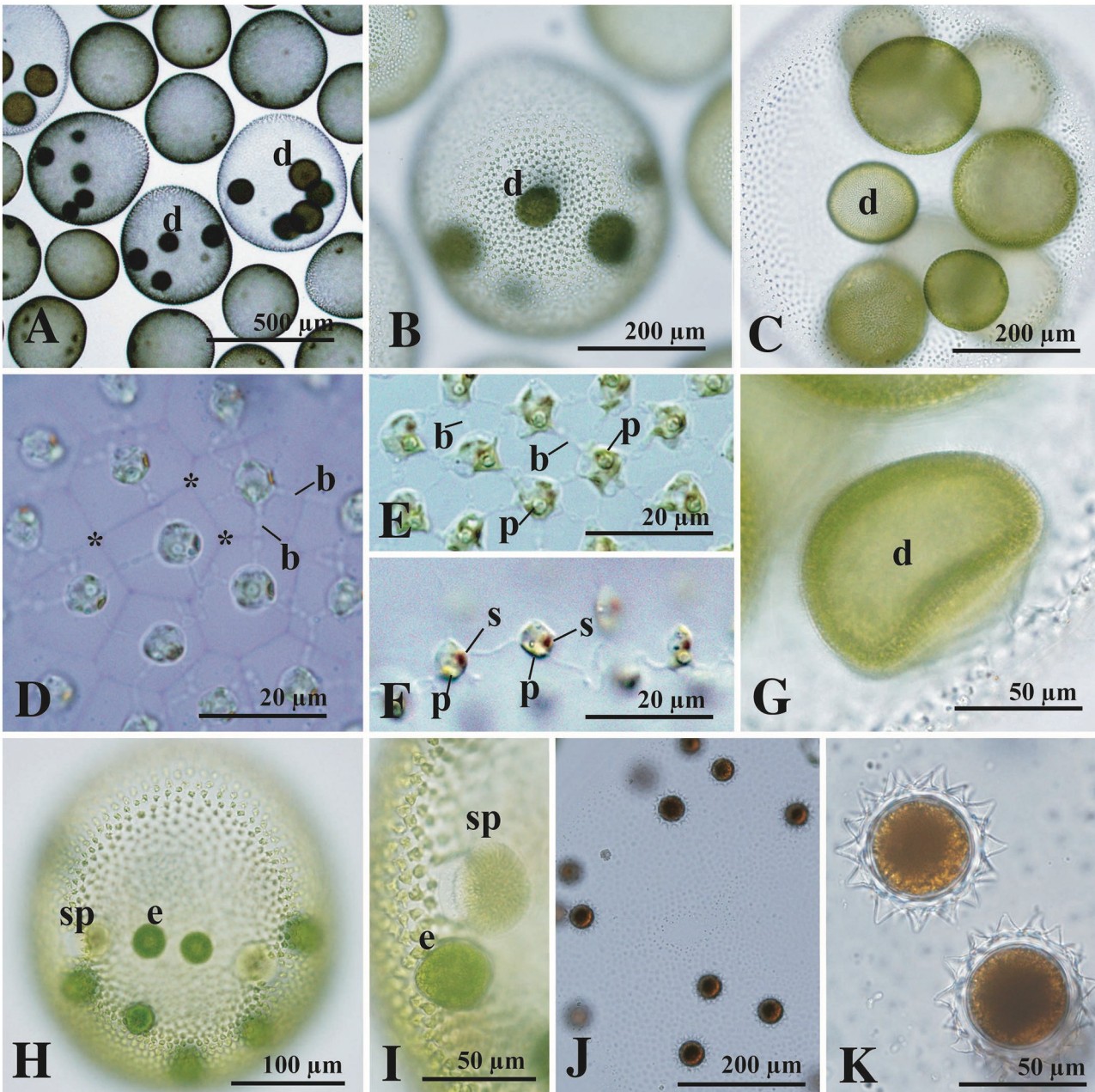

**Fig 4. Light microscopy of *Volvox biwakoensis* sp. nov. strain 2022-1021-Biwa1 from Lake Biwa.** (A–D, G–K) Bright-field microscopy. (E, F) Nomarski differential interference contrast microscopy. (A–C) Asexual spheroids with developing embryos or daughter spheroids (d). (D) Part of asexual spheroid showing individual sheaths (asterisks) and thick cytoplasmic bridges (b) between somatic cells. Stained with methylene blue. (E) Transverse section of somatic cells in asexual spheroid showing cytoplasmic bridges (b) and pyrenoid (p) in the chloroplast. (F) Side view of anterior somatic cells showing stigma (s) and pyrenoid (p) in the chloroplast. (G) Compact daughter spheroid (d) just after inversion. Note that differentiation of gonidia of the next generation is not evident. (H–K) Sexual reproduction. (H) Sexual spheroid with eggs (e) and sperm packets (sp). (I) Part of sexual spheroid showing egg (e) and sperm packet (sp). (J) Sexual spheroid with matured zygotes. (K) Matured zygotes with acute spines on zygote walls.

### *Volvox kirkiorum* (Fig 5)

The strains of *V. kirkiorum* described here from Lake Biwa had essentially the same morphological features as the original description of this species (NIES-543, NIES-2740 [19]), especially in zygote spine morphology and zygote size, and occasional production of small sexual

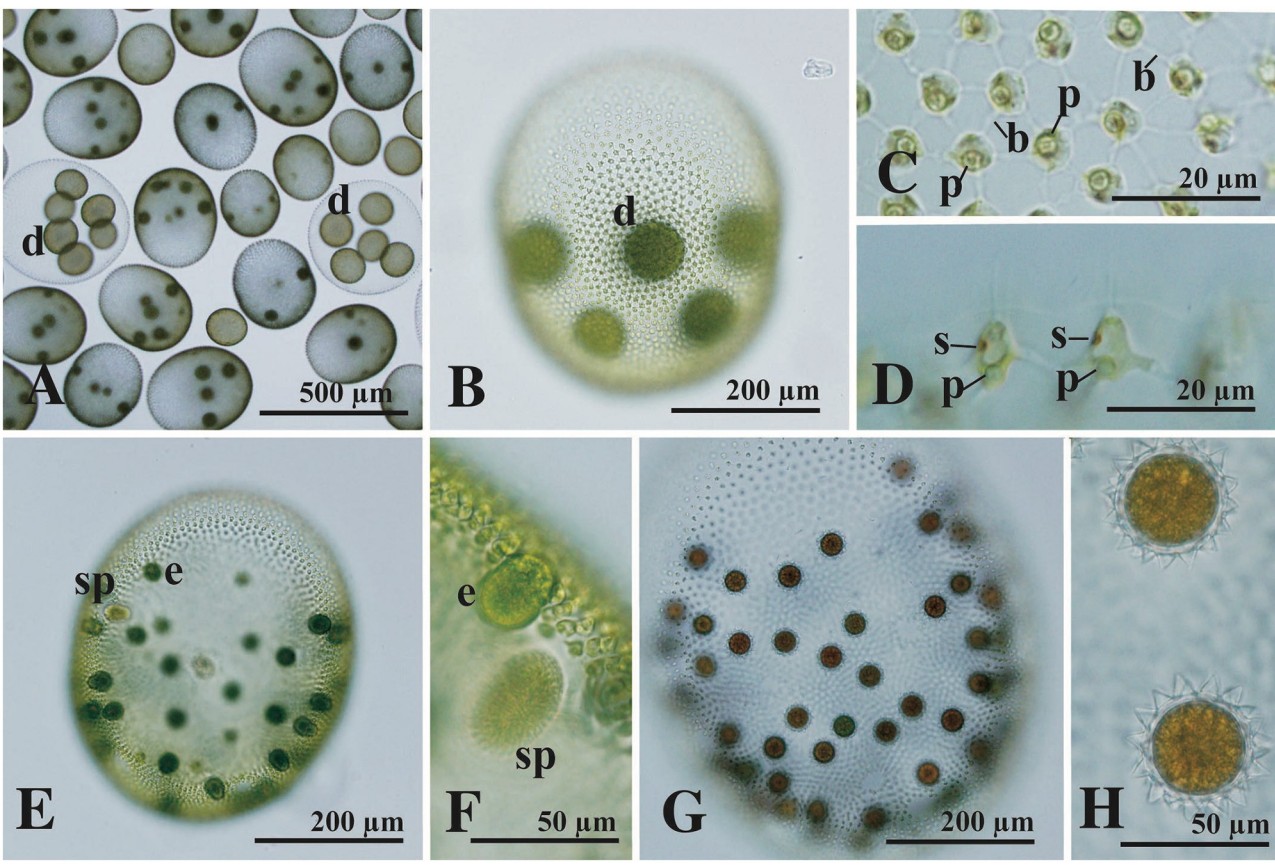

**Fig 5.** Bright-field microscopy of *Volvox kirkiorum* strains 2022-1027-VVx9 (A, C, D) and 2022-1027-VVx11 (B, E-H) from Lake Biwa. (A, B) Asexual spheroids with developing embryos or daughter spheroids (d). (C, D) Part of asexual spheroids showing somatic cells. (C) Transverse section of somatic cells showing cytoplasmic bridges (b) and pyrenoid (p) in the chloroplast. (D) Side view of anterior somatic cells showing stigma (s) and pyrenoid (p) in the chloroplast. (E–H) Sexual reproduction. (E) Sexual spheroid with eggs (e) and sperm packets (sp). (F) Part of sexual spheroid showing egg (e) and sperm packet (sp). (G) Sexual spheroid with matured zygotes. (H) Matured zygotes with acute spines on zygote walls.

spheroids with up to 10 eggs and 4–8 sperm packets (S2A Fig). However, the strains from Lake Biwa sometimes produced greater numbers of zygotes in a sexual spheroid than the original description [18]. More than 90 zygotes were counted in some sexual spheroids of the Lake Biwa strain (S2B Fig), whereas the original description of *V. kirkiorum* indicated up 80 zygotes in a sexual spheroid [19]. However, no genetic differences were found between strains of *V. kirkiorum* (Table 1). Although genetic differences were detected in the *actin* gene sequence between strains of *Volvox* sp. Sagami (Fig 2), sequences of *actin* genes and three other DNA regions examined in the present study were identical among the four Japanese strains of *V. kirkiorum*, i.e., NIES-543, NIES-2740, and Lake Biwa strains 2022-1027-VVx9 and 2022-1027-VVx11 (Fig 2).

## Conclusions

Based on the results of metabarcoding analysis targeting the ITS-2 rDNA regions from Lake Biwa samples [25], 10 unclassified sequences could be considered to be derived from *Volvox* sect. *Volvox* or closely related lineages (S3 Fig and S3 Appendix). These sequences were subdivided into three groups based on phylogenetic analysis in the present study. One of the three

was closely related to *Volvox* sp. Sagami, while the second (four different sequences) formed a monophyletic group with *V. kirkiorum* and the third may represent an unknown basal lineage within or just outside the clade of *Volvox* sect. *Volvox*. *V. kirkiorum* is also distributed outside Lake Biwa in Japan (Table 1) [19]. Therefore, it may have formed a species flock within Lake Biwa and only a single genotype within this lineage may have been transmitted to other regions of Japan (Table 1) although some sequence errors may have affected the apparent genetic divergence within the lineage (S3 Fig). In contrast, ITS-2 rDNA sequences closely related to *V. biwakoensis* were not recognized in ITS-2 rDNA metabarcoding sequences from Lake Biwa (S3 Fig) [25]. Therefore, *V. biwakoensis* may be very rare in Lake Biwa. This new species exhibited genetic separation from its related species based on the ITS secondary structure (Fig 3), and the results of phylogenetic analyses of a combined data set from the nuclear gene *actin*, ITS, and two plastid genes showed that it represented a new phylogenetic lineage within *Volvox* sect. *Volvox* (Fig 2). Therefore, *V. biwakoensis* may represent possible endemism in Lake Biwa. This endemic occurrence of *V. biwakoensis* may reflect the geological history of Lake Biwa. Lake Biwa has a geological history of about four million years [13], while the time of divergence between *V. rousseletii* and *V. ferrisii* (Fig 2) has been estimated to be about five million years ago [26]. Since the present phylogenetic tree suggested that the *V. biwakoensis* lineage diverged before the divergence between *V. rousseletii* and *V. ferrisii* (Fig 2), *V. biwakoensis* may have been endemic to Lake Biwa from the early time of Lake Biwa. However, detailed and reliable taxonomic information on the *Volvox* sect. *Volvox* is very limited within freshwater habitats worldwide, including ancient lakes (S2 Table). Further studies, especially studies using Asian samples of *Volvox* sect. *Volvox*, are required to evaluate the endemism hypothesis of *V. biwakoensis*.

## Taxonomic treatment

*Volvox biwakoensis* Nozaki & H. Yamaguchi sp. nov.   Asexual spheroids ovoid with a narrow anterior face, with usually 5–8 developing embryos or daughter spheroids in the posterior three fifths of the parental spheroid, and composed of 6000–9000 cells embedded in individual sheaths of the gelatinous matrix. The mature asexual spheroid measuring up to 800 μm in length. Somatic cells having two equal flagella, a cup-shaped chloroplast with a single basal pyrenoid and a single eyespot, measuring up to 8 μm in length. Cells connected by thick cytoplasmic bridges. Anterior somatic cells ovoid to ellipsoidal with a narrow anterior face; the cell length greater than or nearly equivalent to the cell width. Reproductive cells or gonidia of the next generation not evident until the inversion stage. Sexual spheroids 3600–5000-celled, bisexual or monoicous with 4–26 eggs and 4–8 sperm packets distributed randomly within the posterior two-thirds of the spheroid, measuring up to 650 μm in length. Mature zygotes with a spiny cell wall, measuring 46–50 μm in diameter (excluding spines). Fully developed spines of zygote wall 5–7 μm in length, straight or slightly curved with an acute apex.

Holotype: Fig 4H showing eggs and sperm packets in bisexual spheroid of strain 2022-1021-Biwa1. This strain has been deposited to the Microbial Culture Collection at the National Institute for Environmental Studies, Japan [15], and available as NIES-4661. The holotype of *V. biwakoensis* is an effectively published illustration since it is impossible to preserve a specimen that would show the features attributed to the taxon, especially the bisexual sexual spheroid (Article 40.5 of International Code of Nomenclature for algae, fungi, and plants [23]).

Strains examined: 2022-1021-Biwa1 (= NIES-4661) (Table 1).

Etymology: The species epithet "*biwakoensis*" meaning "from Lake Biwa (Japanese name "Biwako")".

Type locality: An offshore from Yanagasaki in the south basin (Nanko) of Lake Biwa, Japan (35.0261274, 135.8673386). A water samples was collected by KS on 20 October 2022.

## Supporting information

**S1 Table. Oligonucleotide primers used for amplifying and sequencing *actin* genes.**
(XLSX)

**S2 Table. Comparison of *Volvox biwakoensis* sp. nov. and previously described bisexual/ monoicous morphological type and species of *Volvox* sect. *Volvox*.**
(DOCX)

**S1 Fig. Bright-field microscopy of *Volvox* sp. Sagami strain SB01 from Tokyo, Japan.**
(DOCX)

**S2 Fig. Bright-field microscopy of *Volvox kirkiorum* from Lake Biwa, Japan.**
(DOCX)

**S3 Fig. Phylogeny of *Volvox* sect. *Volvox* and its related metabarcoding sequences (blue) based on the Internal Transcribed Spacer (ITS) regions of nuclear ribosomal DNA (rDNA) (ITS-1, 5.8S rDNA, and ITS-2).**
(DOCX)

**S1 Appendix. Alignment used for phylogeny in Fig 1.**
(TXT)

**S2 Appendix. Alignment used for phylogeny in Fig 2.**
(TXT)

**S3 Appendix. Alignment used for phylogeny in S3 Fig.**
(TXT)

## Acknowledgments

We are grateful to staffs of MCC-NIES who helped our experimental works.

## Author Contributions

**Conceptualization:** Hisayoshi Nozaki, Noriko Ueki, Wirawan Heman, Haruyo Yamaguchi.

**Data curation:** Hisayoshi Nozaki, Ryo Matsuzaki.

**Formal analysis:** Hisayoshi Nozaki, Ryo Matsuzaki.

**Funding acquisition:** Hisayoshi Nozaki, Koichi Shimotori, Noriko Ueki.

**Investigation:** Hisayoshi Nozaki, Ryo Matsuzaki, Koichi Shimotori, Noriko Ueki, Haruyo Yamaguchi, Masanobu Kawachi.

**Methodology:** Hisayoshi Nozaki, Ryo Matsuzaki, Koichi Shimotori, Wirawan Heman, Wuttipong Mahakham.

**Project administration:** Hisayoshi Nozaki, Wuttipong Mahakham, Haruyo Yamaguchi, Yuuhiko Tanabe, Masanobu Kawachi.

**Resources:** Koichi Shimotori, Noriko Ueki, Wirawan Heman, Wuttipong Mahakham, Haruyo Yamaguchi, Yuuhiko Tanabe, Masanobu Kawachi.

**Supervision:** Hisayoshi Nozaki, Koichi Shimotori, Wuttipong Mahakham, Haruyo Yamaguchi, Yuuhiko Tanabe, Masanobu Kawachi.

**Visualization:** Hisayoshi Nozaki, Ryo Matsuzaki.

**Writing – original draft:** Hisayoshi Nozaki, Ryo Matsuzaki, Koichi Shimotori, Noriko Ueki, Haruyo Yamaguchi, Yuuhiko Tanabe.

**Writing – review & editing:** Hisayoshi Nozaki, Ryo Matsuzaki, Koichi Shimotori, Noriko Ueki, Wirawan Heman, Wuttipong Mahakham, Haruyo Yamaguchi, Yuuhiko Tanabe, Masanobu Kawachi.

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
