## [Decision Letter · Decision Letter 0]

9 Jul 2024

PONE-D-24-23039

Two species of the green algae Volvox sect. Volvox from the Japanese ancient lake, Lake Biwa

PLOS ONE

Dear Dr. Nozaki,

Thank you for submitting your manuscript to PLOS ONE. After careful consideration, we feel that it has merit but does not fully meet PLOS ONE’s publication criteria as it currently stands. Therefore, we invite you to submit a revised version of the manuscript that addresses the points raised during the review process.

We look forward to receiving your revised manuscript.

Kind regards,

Michael Klymkowsky, Ph.D.

Academic Editor

PLOS ONE

Journal Requirements:

2. "We noticed you have some minor occurrence of overlapping text with the following previous publication(s), which needs to be addressed:

https://doi.org/10.1371/journal.pone.0235622

In your revision ensure you cite all your sources (including your own works), and quote or rephrase any duplicated text outside the methods section. Further consideration is dependent on these concerns being addressed."

Additional Editor Comments:

**Both reviewers call for minor revisions that expect with be simple to address. **

Reviewers' comments:

Reviewer's Responses to Questions

**Comments to the Author**

1. Is the manuscript technically sound, and do the data support the conclusions?

Reviewer #1: Yes

Reviewer #2: Yes

2. Has the statistical analysis been performed appropriately and rigorously? 

Reviewer #1: Yes

Reviewer #2: N/A

3. Have the authors made all data underlying the findings in their manuscript fully available?

Reviewer #1: Yes

Reviewer #2: Yes

4. Is the manuscript presented in an intelligible fashion and written in standard English?

Reviewer #1: Yes

Reviewer #2: Yes

5. Review Comments to the Author

Reviewer #1: The manuscript "Two species of the green algae Volvox sect. Volvox from the Japanese ancient lake, Lake Biwa" by Nozaki et al. identifies two strains of Volvox sect. Volvox including a new species from Lake Biwa as an ancient lake, which is potentially an important source of ecology and biodiversity studies but lacks the taxonomic identification of Volvox sec. Volvox at the species level worldwide. Therefore the new species Volvox biwakoensis sp. nov. with distinct zygote morphology and number among the related species should shed light on the natural history of aquatic microorganisms in ancient lakes. The authors clearly and fully described the strains they established for the publication.

Major comment

In the introduction, the manuscript discusses species diversity in ancient lakes and presents the discovery of a new species in one such lake. However, the subsequent sections do not adequately address how this discovery contributes to our understanding of species diversity in ancient lakes. Does the "possible endemism" of Volvox biwakoensis (p.15 l.320) reflect the geological history of Lake Biwa?

Minor comments

p.4 ll.95-6

"the south basin (Nanko) of Lake Biwa" is appropriate because Nanko is not a separate lake from Lake Biwa.

p.12 l.257

Was germination attempted without success or it was not attempted?

p.15 l.331

"thick" instead of "chick"

Reviewer #2: I support the publication of this article after minor revision. In terms of its content and illustrations (figures and tables), the article is at a very high scientific level. I only have two critical comments.

1. It is advisable to provide additional information about the ancient Lake Biwa: the surface area of the lake, its average and maximum depth and the elevation of the surface of the lake above sea level.

2. The second critical comment concerns supporting information: in S2 Table (Comparison of Volvox biwakoensis sp. nov. and previously described bisexual/monoicous morphological type and species of Volvox sect. Volvox) the data presented on the geographical distribution of several species of Volvox are incomplete. It is necessary to add information obtained by Mary Pocock (the recognized expert in this field) about Volvox in Australia and New Zealand:

Volvox merrillii is known not only from the Philippines and India (as listed in the S2 Table), but also from Australia (New South Wales) - M.S. Cave & M.A. Pocock. Karyological Studies in the Volvocaceae. American Journal of Botany, Vol. 38, No. 10 (Dec., 1951), pp. 800-811.

Regarding the distribution of Volvox barberi and Volvox globator, it should be added that Pocock found these two species in New Zealand - Pocock M.A. Notes on the occurrence in New Zealand of Volvulina steinii Pleyfair and species of Volvox Linn. Rec. Cant. Mus., 6: 1. This hard-to-find article is cited in: Chapman VJ, Thompson RH, Segar ECM (1957) Check list of fresh-water algae of New Zealand. Trans. Roy. Soc. New Zealand. 84(4): 695–747.

6. PLOS authors have the option to publish the peer review history of their article (what does this mean?). If published, this will include your full peer review and any attached files.

Reviewer #1: No

Reviewer #2: No

---

## [Author Response · Author response to Decision Letter 0]

12 Jul 2024

Dear Dr. Michael Klymkowsky, 

Thank you very much for your scholarly and positive decision of our manuscript entitled " Two species of the green algae Volvox sect. Volvox from the Japanese ancient lake, Lake Biwa" for publication in PLOS ONE as a research article.

 Based on the comments/suggestions raised by the two reviewers, we have revised the manuscript completely. Our responses to the comments have been described in the followings:

Sincerely,

Hisayoshi Nozaki

PONE-D-24-23039

Two species of the green algae Volvox sect. Volvox from the Japanese ancient lake, Lake Biwa

PLOS ONE

Response: These three files have been uploaded. In addition, S2 Table with tracked changes and S3 Figure with tracked changes have also prepared, in addition to the unmarked revised version.

Response: As suggested, all funding-related text has been removed from ‘Financial Disclosure’ sections of the revised manuscript. Figures 1-5 have been revised by using PACE and five tiff files of Figures 1-5 have been uploaded.

We look forward to receiving your revised manuscript.

Kind regards,

Michael Klymkowsky, Ph.D.

Academic Editor

PLOS ONE

Journal Requirements:

Response: I have ensured that our manuscript meets PLOS ONE's style requirements, including those for file naming.

2. "We noticed you have some minor occurrence of overlapping text with the following previous publication(s), which needs to be addressed:

https://doi.org/10.1371/journal.pone.0235622

In your revision ensure you cite all your sources (including your own works), and quote or rephrase any duplicated text outside the methods section. Further consideration is dependent on these concerns being addressed."

Response: In the description of the holotype, the sentences have been changed. In addition, figure legends of Figures 1, 2, and S3 Figure have been revised in explanation of BI and bootstrap values and branch lengths. However, "Nomenclature" has not been changed because it belongs to the methods section. In S2 Table, Nozaki et al. [1] (https://doi.org/10.1371/journal.pone.0235622) was already cited in the top of the table.

Response: All funding-related text has been removed from the revised main manuscript.

Response: As described above, all funding-related text has been removed from the main manuscript of revised version. 

The correct grant numbers for the awards we received for our study have been described in the ‘Funding Information’

 as follows:

JSPS KAKENHI

Award Number: 19K22446, 24K08946 | Recipient: Hisayoshi Nozaki, Ph.D.

JSPS KAKENHI

Award Number: 21K06295 | Recipient: Noriko Ueki, Ph.D.

The Collaborative Research Fund from Shiga Prefecture entitled “Study on water quality and lake-bottom environment for protection of the soundness of water environment” under the Japanese Grant for Regional Revitalization

Award Number: OS2024RR1 | Recipient: Koichi Shimotori, Ph.D.

Network Joint Research Center for Materials and Devices

Award Number: 20211204, 20221238, 20231201, 20241233 | Recipient: Noriko Ueki, Ph.D.

We prepared Financial Disclosure as suggested*. However, because of the possible error of the editorial manager system of PLOS ONE, we cannot access 'Financial Disclosure' of Additional Information via the editorial manager system. 

Please help me!

*This study was partially supported by a Grants-in-Aid for Scientific Research (grant numbers 19K22446 and 24K08946 for HN, 21K06295 to NU) from the Ministry of Education, Culture, Sports, Science and Technology (MEXT)/Japan Society for the Promotion of Science (JSPS) KAKENHI (https://www.jsps.go.jp/english/e-grants/), the Collaborative Research Fund (grant number OS2024RR1 to KS) from Shiga Prefecture entitled “Study on water quality and lake-bottom environment for protection of the soundness of water environment” under the Japanese Grant for Regional Revitalization (https://www.mlit.go.jp/en/index.html), and the Research Program of "Dynamic Alliance for Open Innovation Bridging Human, Environment and Materials" (grand numbers 20211204, 20221238, 20231201 and 20241233 for NU) in "Network Joint Research Center for Materials and Devices" (https://alliance.tagen.tohoku.ac.jp/english/). The funders had no role in study design, data collection and analysis, decision to publish, or preparation of the manuscript.

Response: I have asked DDBJ to release the new sequence data immediately. The DDBJ/ENA/GenBank accession numbers (LCLC817957–LC817993 and LC818981) are now available.

Response: A book [13] has been added to reference the details of Lake Biwa (see comment 1 of Reviewer #2) and the divergence has been discussed based on the major comment by reviewer#1 by citing [26]. :

13. Kawanabe H, Nishino M, Maehata M, editors. Lake Biwa: interactions between nature and people. New York: Springer; 2012. doi: 10.1007/978-94-007-1783-1

26. Yamagishi S, Yamamoto K, Takahashi K, Kawai-Toyooka H, Suzuki S, Matsuzaki R, Yamaguchi H, Kawachi M, Higashiyama T, Nozaki H. Evolutionary analysis of MID homologs during the transition from homothallic species to heterothallic species in Volvox sect. Volvox (Chlorophyceae). Phycol. Res. 2024; 72: 46-55. doi: 10.1111/pre.12538

Additional Editor Comments:

Both reviewers call for minor revisions that expect with be simple to address. 

Reviewers' comments:

Reviewer's Responses to Questions

Comments to the Author

1. Is the manuscript technically sound, and do the data support the conclusions?

Reviewer #1: Yes

Reviewer #2: Yes

2. Has the statistical analysis been performed appropriately and rigorously?

Reviewer #1: Yes

Reviewer #2: N/A

3. Have the authors made all data underlying the findings in their manuscript fully available?

Reviewer #1: Yes

Reviewer #2: Yes

4. Is the manuscript presented in an intelligible fashion and written in standard English?

Reviewer #1: Yes

Reviewer #2: Yes

5. Review Comments to the Author

Reviewer #1: The manuscript "Two species of the green algae Volvox sect. Volvox from the Japanese ancient lake, Lake Biwa" by Nozaki et al. identifies two strains of Volvox sect. Volvox including a new species from Lake Biwa as an ancient lake, which is potentially an important source of ecology and biodiversity studies but lacks the taxonomic identification of Volvox sec. Volvox at the species level worldwide. Therefore the new species Volvox biwakoensis sp. nov. with distinct zygote morphology and number among the related species should shed light on the natural history of aquatic microorganisms in ancient lakes. The authors clearly and fully described the strains they established for the publication.

Major comment

In the introduction, the manuscript discusses species diversity in ancient lakes and presents the discovery of a new species in one such lake. However, the subsequent sections do not adequately address how this discovery contributes to our understanding of species diversity in ancient lakes. Does the "possible endemism" of Volvox biwakoensis (p.15 l.320) reflect the geological history of Lake Biwa?

Response: Yes, the "possible endemism" of Volvox biwakoensis (p.15 l.320) may reflect the geological history of Lake Biwa. Thus, several sentences of discussion have been added to the conclusion section in the revised manuscript.

Minor comments

p.4 ll.95-6

"the south basin (Nanko) of Lake Biwa" is appropriate because Nanko is not a separate lake from Lake Biwa.

Response: Revised as suggested. Thank you!

p.12 l.257

Was germination attempted without success or it was not attempted?

Response: It was not attempted because of lack of abundant matured zygotes for induction of zygote germination. The description of the text has been revised: "Induction of zygote germination was not attempted."

p.15 l.331

"thick" instead of "chick"

Response: Revised as suggested. Thank you!

Reviewer #2: I support the publication of this article after minor revision. In terms of its content and illustrations (figures and tables), the article is at a very high scientific level. I only have two critical comments.

1. It is advisable to provide additional information about the ancient Lake Biwa: the surface area of the lake, its average and maximum depth and the elevation of the surface of the lake above sea level.

Response: These data have been added to the Method section of the revised manuscript.

2. The second critical comment concerns supporting information: in S2 Table (Comparison of Volvox biwakoensis sp. nov. and previously described bisexual/monoicous morphological type and species of Volvox sect. Volvox) the data presented on the geographical distribution of several species of Volvox are incomplete. It is necessary to add information obtained by Mary Pocock (the recognized expert in this field) about Volvox in Australia and New Zealand:

Volvox merrillii is known not only from the Philippines and India (as listed in the S2 Table), but also from Australia (New South Wales) - M.S. Cave & M.A. Pocock. Karyological Studies in the Volvocaceae. American Journal of Botany, Vol. 38, No. 10 (Dec., 1951), pp. 800-811.

Regarding the distribution of Volvox barberi and Volvox globator, it should be added that Pocock found these two species in New Zealand - Pocock M.A. Notes on the occurrence in New Zealand of Volvulina steinii Pleyfair and species of Volvox Linn. Rec. Cant. Mus., 6: 1. This hard-to-find article is cited in: Chapman VJ, Thompson RH, Segar ECM (1957) Check list of fresh-water algae of New Zealand. Trans. Roy. Soc. New Zealand. 84(4): 695–747.

Responses: Thank you very much for your very valuable and helpful information of distribution of Volvox sect. Volvox. These data have been described in S2Table by citing the two references, Pocock (1951), Cave & Pocock (1951).

---

## [Editor Report · Decision Letter 1]

3 Sep 2024

Two species of the green algae Volvox sect. Volvox from the Japanese ancient lake, Lake Biwa

PONE-D-24-23039R1

Dear Dr. Nozaki,

We’re pleased to inform you that your manuscript has been judged scientifically suitable for publication and will be formally accepted for publication once it meets all outstanding technical requirements.

Kind regards,

Abul Khayer Mohammad Golam Sarwar

Academic Editor

PLOS ONE
---

## [Editor Report · Acceptance letter]

13 Sep 2024

PONE-D-24-23039R1 

PLOS ONE

Dear Dr. Nozaki, 

I'm pleased to inform you that your manuscript has been deemed suitable for publication in PLOS ONE. Congratulations! Your manuscript is now being handed over to our production team.

Kind regards, 

on behalf of

Professor Abul Khayer Mohammad Golam Sarwar 

Academic Editor

PLOS ONE